# Assessment of the Dissimilarities of EDI and SPI Measures for Drought Determination in South Africa

**Omolola M. Adisa** [1,*], **Muthoni Masinde** [1] **and Joel O. Botai** [1,2]

1    Department of Information Technology, Central University of Technology, Free State, Private Bag X200539, Bloemfontein 9300, South Africa; emasinde@cut.ac.za (M.M.); joel.botai@weathersa.co.za (J.O.B.)
2    South African Weather Service, Private Bag X097, Pretoria 0001, South Africa
*    Correspondence: lolaadisa@yahoo.com; Tel.: +27-84-849-1170

**Abstract:** This study examines the (dis)similarity of two commonly used indices Standardized Precipitation Index (SPI) computed over accumulation periods 1-month, 3-month, 6-month, and 12-month (hereafter SPI-1, SPI-3, SPI-6, and SPI-12, respectively) and Effective Drought Index (EDI). The analysis is based on two drought monitoring indicators (derived from SPI and EDI), namely, the Drought Duration (DD) and Drought Severity (DS) across the 93 South African Weather Service's delineated rainfall districts over South Africa from 1980 to 2019. In the study, the Pearson correlation coefficient dissimilarity and periodogram dissimilarity estimates were used. The results indicate a positive correlation for the Pearson correlation coefficient dissimilarity and a positive value for periodogram of dissimilarity in both the DD and DS. With the Pearson correlation coefficient dissimilarity, the study demonstrates that the values of the SPI-1/EDI pair and the SPI-3/EDI pair exhibit the highest similar values for DD, while the SPI-6/EDI pair shows the highest similar values for DS. Moreover, dissimilarities are more obvious in SPI-12/EDI pair for DD and DS. When a periodogram of dissimilarity is used, the values of the SPI-1/EDI pair and SPI-6/EDI pair exhibit the highest similar values for DD, while SPI-1/EDI displayed the highest similar values for DS. Overall, the two measures show that the highest similarity is obtained in the SPI-1/EDI pair for DS. The results obtainable in this study contribute towards an in-depth knowledge of deviation between the EDI and SPI values for South Africa, depicting that these two drought indices values are replaceable in some rainfall districts of South Africa for drought monitoring and prediction, and this is a step towards the selection of the appropriate drought indices.

**Keywords:** drought; comparison; Effective Drought Index (EDI); Standard Precipitation Index (SPI)





## 1. Introduction

South Africa's water resources, food security, infrastructure, health, as well as its ecosystem facilities, and biodiversity are threatened by climate change [1]. Two prominent natural disasters, namely, drought and flood, occur in South Africa. Drought is an unusually dry condition continuous over a long period [2]. Droughts, as well as floods, are associated with strong and severe weather events. While the impacts of floods are instantly noticeable, drought impacts are assessed over a period of time. The impacts are spread across several sectors of the economy, such as agriculture, water, tourism, transport, energy, and ecosystem [3]. These impacts are associated with the death of livestock, rivers, and reservoirs drying up, crops wilt, and socio-economic loss [4]. Drought can be categorized by its severity, duration, and areal extent. Several distressing drought events have occurred in the past few decades. Among these is the severe drought that occurred in 2009, and affected diverse parts of the world, but more people were affected in Africa than in other places [5]. Recently (2016–2018), South Africa has experienced a lengthy drought period [6,7] that has affected both water resources and agricultural production, with the effects already propagated into the socio-economic systems. This drought event

is regarded as one of the worst droughts in the country's history, leading to the death of many and millions forced into further hardship [7]. Natural drought events abound, but with monitoring, prediction, and early warning, drought preparedness could be enhanced, impacts can be reduced, and adaptive measures put in place.

According to Wilhite et al. [8,9], drought is categorized into four: (a) Meteorological, (b) agricultural, (c) hydrological, and (d) socio-economic. Meteorological drought is triggered by a prolonged shortage of precipitation from its long-term mean [8]. On the other hand, agricultural drought is characterized by shortages of total soil moisture and results primarily from the deficit of precipitation, whilst hydrological drought is associated with the persistent scarcity or absence of water in water reservoirs, aquifers, or courses [8–14]. Furthermore, socio-economic drought conditions result from the negative effects of meteorological, agricultural, and hydrological droughts on the socio-economic sectors. Drought occurrence is a result of some environmental factors, among which are rainfall intensity, duration and severity, temperature, relative humidity, and wind flow [15]. Recently, the attention of several researchers has been drawn to drought studies because there has been a significant increase in the incidence, intensity, and area affected by drought. This is primarily instigated by the activities of humans, as well as the effect of climate change [16]. It is of utmost importance to understand drought characteristics at a regional level in a quest to alleviate drought risk, moderate latent effects on innumerable socio-economic sectors, and implement appropriate procedures and policies [17,18].

Drought indices are widely used for drought monitoring, such as Palmer Drought Severity Index (PDSI; [19]), Rainfall Decile based Drought Index (RDDI; [20]), Crop Moisture Index (CMI; [21]), Bhalme and Mooley Drought Index (BMDI; [22]), Surface Water Supply Index (SWSI; [23]), Standardized Precipitation Index (SPI; [24]), Soil Moisture Drought Index (SMDI; [25]), Effective Drought Index (EDI; [26]), China Z-Index (CZI; [27]), Soil Moisture Deficit Index (SMDI; [13]), Reconnaissance Drought Index (RDI; [28]), Standardized Precipitation Evapotranspiration Index (SPEI; [29]) Agricultural Reference Index for Drought (ARID; [30]), the Vegetation Health Index (VHI; [31]). The majority of these indices are based on incessant functions of one or more of these hydro-meteorological variables like temperature, precipitation, potential evapotranspiration, soil water, groundwater, run-off, streamflow (World Meteorological Organization (WMO) [32]).

Drought indices are usually region-specific, and they are limited in their applications to diverse climatic conditions, given the intrinsic complexity of drought phenomena [10]. For instance, PDSI is broadly used in the United States, the RDDI is operational in Australia, and the CZI is used by the National Metrological Center in China [33]. Although contrary to Heim [10], Dai [34] reveals the global applicability of PDSI, in regions of extreme elevation and high geographic latitudes. The SPI is one of the most popularly used in the investigation of meteorological drought. There may be a disparity in the complexity of drought, the existence of good quality data, and the efficacy of drought indices in depicting historical drought events for different locations [33]. It is of paramount importance to identify a suitable drought index for a particular region to quantify and prepare for drought-related disasters. Several comparative studies of drought indices have been conducted in several regions. For instance, Ref. [33] compared six drought indices to detect the most suitable for drought monitoring in the Ken River basin, India, and specified that EDI is a more appropriate drought index for the study basin. In Ref. [35], the suitability of seven drought indices for drought monitoring in the basin was compared and suggested the use of EDI and SPI. Their study also specified that the EDI was more receptive to drought and performed better compared to the SPI. In Ref. [36], 14 drought indices in two areas of the United States (US) were appraised and ranked and concluded that deciles and SPI ranked best amid the appraised indices. The performance of three drought indices in the Upper Niger basin was examined based on six verdict criteria [37]. They stated that the SPI ranked first amongst meteorological drought indices.

EDI and SPI values are both standardized, this compares the drought severity at two or more locations irrespective of their climatic differences [38]. The EDI has thresholds sig-

nifying the range of wetness from extreme drought conditions to extremely wet conditions like the SPI [39]. The computation of SPI encompasses an analysis of the monthly rainfall deviation from its recorded monthly rainfall series whereas, EDI computation involves the analysis of the daily rainfall deviation from its recorded daily series. Ref. [40] modified the algorithm to permit the EDI application to accommodate monthly data. Furthermore, the SPI is a drought index that has been used extensively in various parts of the world, while the EDI is a fairly new drought index, and its applicability has not been tested extensively [35]. Additionally, the WMO suggested using SPI as the core meteorological drought index countries should employ in monitoring and tracking drought conditions [4,41] illustrated the robustness of EDI for the detection of early drought and its similarity with SPI as against other indices.

The growth in the development of drought indices is remarkable. This could be due to the need to adequately monitor and respond to the increasing occurrence of drought. Consequently, there is a growing concern to investigate the sensitivity of drought indices to drought characteristics, such as drought onset, cessation, duration, and severity. The usefulness of such a study has been demonstrated in Ethiopia [39] and Serbia [4] over river basins. Whilst previous studies have investigated drought in South Africa using indices that include SPEI, SPI, and EDI [6,7]; however, as reported by ref. [42,43] there is insufficient knowledge on the sensitivity of the drought monitoring indices across the national landscape. Besides, there is growing interest among the drought research community for operationalizing an integrated drought early warning system for South Africa. Consequently, to ensure the effectiveness of the system, there is a need to understand the advantages and disadvantages, the similarity and dissimilarity of the several drought indices that have been used. Therefore, this study aimed at investigating the (dis)similarity of two commonly used indices (SPI and EDI) based on DD and DS.

## 2. Materials and Methods

### 2.1. Study Area

The study area includes the nine provinces of South Africa, namely, Eastern Cape (EC), Free State (FS), Gauteng (GT), KwaZulu-Natal (KZN), Limpopo (LP), Mpumalanga (MP), Northern Cape (NC), North West (NW) and Western Cape (WC), see Figure 1. It covers a total area of 1,221,037 km$^2$ with about 58 million population. South Africa receives an annual average total rainfall of about 450 mm and average maximum and minimum temperatures of 27 °C and 18 °C, respectively. The FS, MP, and NW provinces fall within regions that receive less than 600 mm of rainfall per year. The FS province is characterized by chilly winters (ranging from a cold 1 °C to mild 17 °C), plenty of sunshine (15 °C to 32 °C), and summer rains at an average of 500 mm annually. In NW province, there is almost year-round sunshine, with an average rainfall of 400 mm annually. The summer temperature ranges from 22 °C to 34 °C. The NW province is characterized by dry, sunny days and chilly nights during winter (2 °C to 20 °C). In KZN, summer is from December to February with temperature ranging between 23 °C to 33 °C, and winter is from June to August with temperature ranging between 16 °C to 25 °C. The province is characterized by long, hot summers with average annual rainfall ranging between 800 mm. Furthermore, the western part of MP province is much colder during winter and hotter during summer than the other parts of the province. The average annual temperature is about 19 °C, and rainfall is between 500 mm and 800 mm annually.

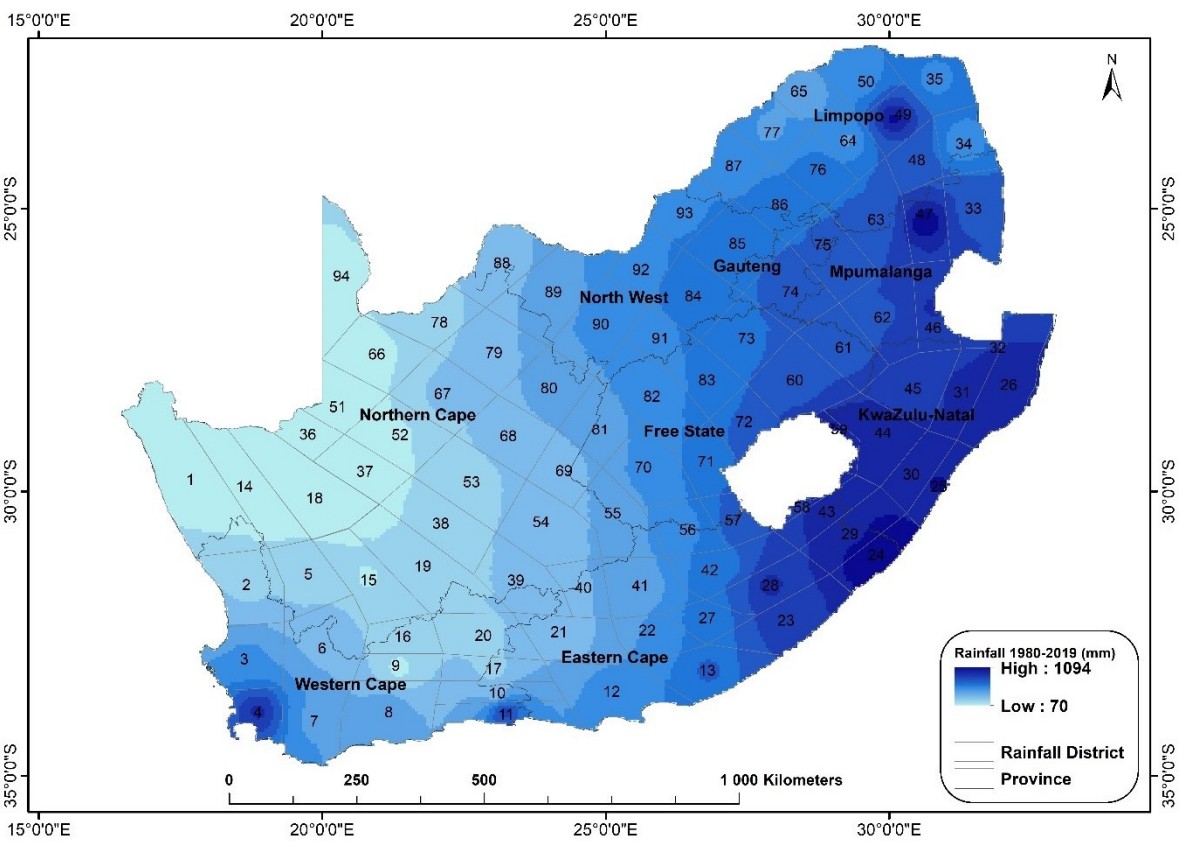

**Figure 1.** Map of rainfall districts for South Africa (SAWB, 1972) with provincial borders showing annual total rainfall 1980–2019.

### 2.2. Materials and Methods

The South African Weather Service (SAWS) monthly rainfall districts data was used as the input data. Detailed information on the delineation of the districts to homogeneous rainfall coverage is reported elsewhere in Reference [2]. To calculate the statistics and interpolation, the centroid of each rainfall district was derived using ArcGIS® software 10.5 by Environmental Systems Research Institute, Inc., California, 2010 (ESRI). The time series of the two drought indices EDI and SPI accumulation periods of 1, 3, 6, and 12 months; were calculated for 93 meteorological stations across South Africa from 1980 to 2019 [24]. Drought duration and severity were calculated based on [6]. Thereafter, the Pearson's correlation coefficient dissimilarity measure with *p*-value and periodogram dissimilarity measure was used to determine the similarity and dissimilarity. The methodological flowchart is shown in Figure 2.

### 2.2.1. Computation of Effective Drought Index (EDI)

According to ref. [26], various existing indices used have limits in indicating the precise start and end of the drought period and the duration of drought. This led to their proposition of EDI, which is another rainfall-related measure, as a remedy for some of these shortcomings. Contrasting to many other drought indices, the EDI in its original form is computed from a daily time step. The EDI (Equation (1)) is a function of the *PRN* (One day's precipitation needed for a return to normal conditions, Equation (2)), this implies the recovery from the accumulated deficit from the commencement of a drought.

$$EDI_j = \frac{PRN_j}{ST\left(PRN_j\right)} \tag{1}$$

$$PRN_j = \frac{DEP_j}{\sum_{N=1}^{j}(1/N)} \tag{2}$$

$$DEP = EP - MEP \tag{3}$$

where $j$ is actual duration, $ST(PRN)$ is the standard deviation of each day's $PRN$, $EP$ is 'effective precipitation', and $MEP$ is the mean of each day's $EP$. The $EP$ (Equation (4)) is the core new concept in the algorithm. The $EP$ denotes to the addition of all daily precipitation with a time decrease function. The $EP$ for any day is a function of precipitation of the present day, as well as of preceding days, but with lower weights. The calculation technique of the EDI commences by applying a dummy water deficit period as a prerequisite for defining the real period. The dummy duration can differ, for instance, it could be 365 days representing the value of the total water resources stored or available for an extended period, or it could be 15 days indicating a short period.

$$EP_i = \sum_{n=1}^{i}\left[\left(\sum_{m=1}^{n} P_m\right)/n\right] \tag{4}$$

where $i$ is the duration of summation and $P_m$ is the precipitation of $m$ days before. For more details, refer to ref. [26].

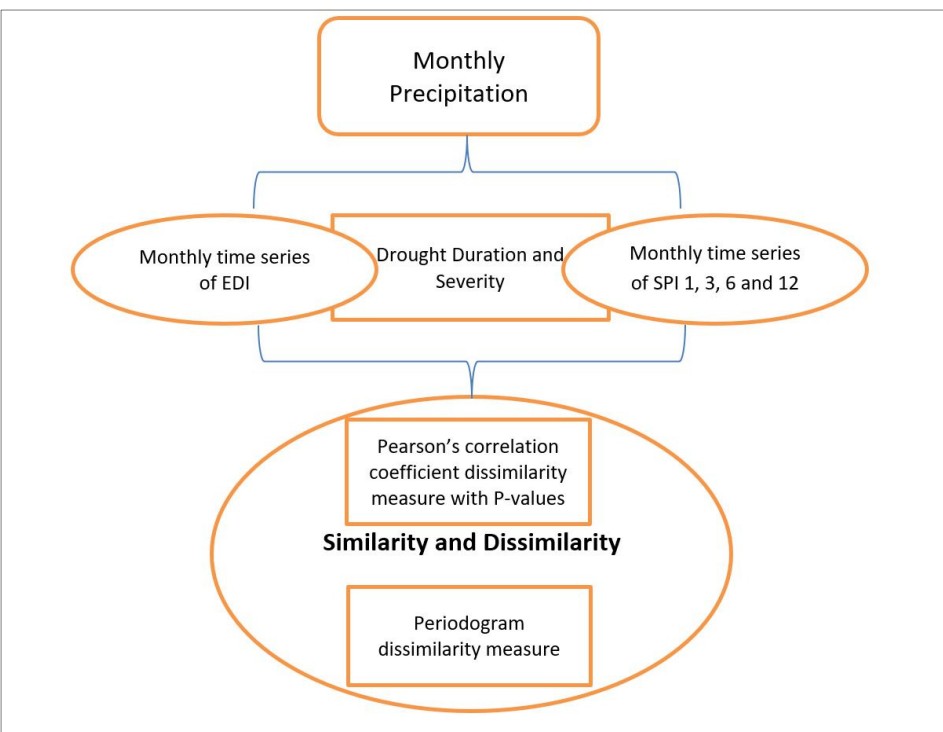

**Figure 2.** Methodological flowchart of assessment of the dissimilarities of EDI and SPI measures for drought determination.

### 2.2.2. Computation of Standardized Precipitation Index (SPI)

To compute the SPI for any location, the long-term precipitation record is required for an anticipated period. The existing long-term rainfall data is fitted to a gamma probability distribution, which further transforms it to a normal distribution to have the SPI mean value at zero [24]. This transformed probability SPI values range from +2.0 to −2.0 (Table 1), with 5% extremes outside this range [44,45]. SPI may be computed at multiple time steps, such as 1, 3, 6, 9, 12, and 24 months; in this study, the 1, 3, 6, and 12-month accumulation period is used. The SPI-$n$ is computed by first fitting precipitation data for each calendar month (or the $n$-month accumulation) to a theoretical distribution [24].

**Table 1.** Different categories of SPI and EDI values for drought severity (Byun and Wilhite (1999)).

| Category | Range of Drought Index Values | |
| --- | --- | --- |
| | SPI | EDI |
| Extremely Dry | $\leq -2.0$ | $\leq -2.0$ |
| Severely Dry | $-1.5$ to $-1.99$ | $-1.5$ to $-1.99$ |
| Moderately Dry | $-1.0$ to $-1.49$ | $-1.0$ to $-1.49$ |
| Normal | $-0.99$ to $0.99$ | $-0.99$ to $0.99$ |
| Moderately Wet | $1.0$ to $1.49$ | $1.0$ to $1.49$ |
| Severely Wet | $1.5$ to $1.99$ | $1.5$ to $1.99$ |
| Extremely Wet | $\geq 2.0$ | $\geq 2.0$ |

### 2.2.3. Methodology for Time Series Classification

A crucial problem in the classification of time series is the choice of an applicable metric [4]. Pearson's correlation coefficient alongside its *p*-value and periodogram dissimilarity measures were used to determine the similarities or dissimilarities between the pair of time series in the data set. The results of these measures were displayed in maps and graphs (Figures 3–10).

### 2.3. Dissimilarity Measure Based on Pearson's Correlation

Correlation is a method for investigating the relationship between two quantitative, continuous variables. Pearson's correlation coefficient is a measure that relates to the strength and direction of a linear relationship. This metric is calculated for the vectors $x$ and $y$ following Equation (6). Pearson's correlation values range from $-1$ to $+1$.

$$CORR(x,y) = \frac{\sum_{i=1}^{n}(x_i - \bar{x})(y_i - \bar{y})}{\sqrt{\sum_{i=1}^{n}(x_i - \bar{x})^2}\sqrt{\sum_{i=1}^{n}(y_i - \bar{y})^2}} \tag{5}$$

Pearson's correlation values range from $-1$ to $+1$. The larger the absolute value of the coefficient, the stronger the relationship between the variables.

### 2.4. Periodogram Based Dissimilarity Measure

Periodogram based dissimilarity [46–48] this measure gives us the Euclidean distance from two-time successions' periodogram coefficients. Since the periodogram variance emerges proportionately to the spectrum value at the equivalent frequencies, using the logarithm of the normalized periodogram is more appropriate. In particular, ref. [48] considers a distance measure concerning the cumulative versions of the periodograms, such as the integrated periodograms. Casado de Lucas claims that the methods relate to the integrated periodogram present quite a few advantages over the ones that related to the periodogram. Particularly, the periodogram is an asymptotically neutral, but unpredictable estimator of the spectral density—whereas, the integrated periodogram is a dependable estimator of the spectral distribution. Theoretically, the spectral distribution constantly occurs, but the spectral density occurs only under categorically continuous distributions. The integrated periodogram wholly controls the stochastic process. The normalized type exact added weight to the shape of the curves compare to the scale, which is precisely what is required in the case of drought indices. Consequently, the normalized logarithm periodogram is computed using Equation (6).

$$d_{LNP}(x,y) = \sqrt{\sum_{j=1}^{[n/2]}\left[logNP_x(w_j) - log\, NP_y(w_j)\right]^2} \tag{6}$$

where $P_x(w_j) = (1/n)\left|\sum_{t=1}^{n} x_t e^{-it\omega j}\right|^2$ and $P_y(w_j) = (1/n)\left|\sum_{t=1}^{n} y_t e^{-it\omega j}\right|^2$ is the periodograms of the time series $x$ and $y$, respectively, at frequency $w_j = 2\pi j/n$, $j = 1, \ldots, [n/2]$ in the range 0 to $\pi$ (where $[n/2]$ is the largest integer less than or equal to $n/2$).

## 3. Results

### 3.1. Comparison of SPI and EDI Using the Pearson Correlation Coefficient and p-Value

The Pearson correlation coefficient estimate dissimilarity between SPI-1, SPI-3, SPI-6, SPI-12, and EDI drought duration (DD) for the SAWS' rainfall districts is portrayed in Figure 3A–D and Figure 4A–D. A two-tailed *T*-test for the Pearson correlation coefficient is calculated over individual rainfall districts, see Figure 3. From Figures 3A and 4A, strong dissimilarity (but statistically insignificant) between SPI-1/EDI pair can be observed in 35% of the rainfall districts in NC province. Furthermore, 73%, 70%, and 60% of SAWS' rainfall districts located in EC, WC, and NW provinces (respectively) exhibit moderate statistically significant dissimilarity, respectively, among the SPI-1/EDI pair. The SPI-1/EDI pair has similar statistically significant values in 90% of the rainfall districts in GT, MP, and KZN provinces. As shown in Figures 3B and 4B, about 10% of the districts located in LP and NC provinces exhibit strong statistically insignificant dissimilarity for the SPI-3/EDI pair values.

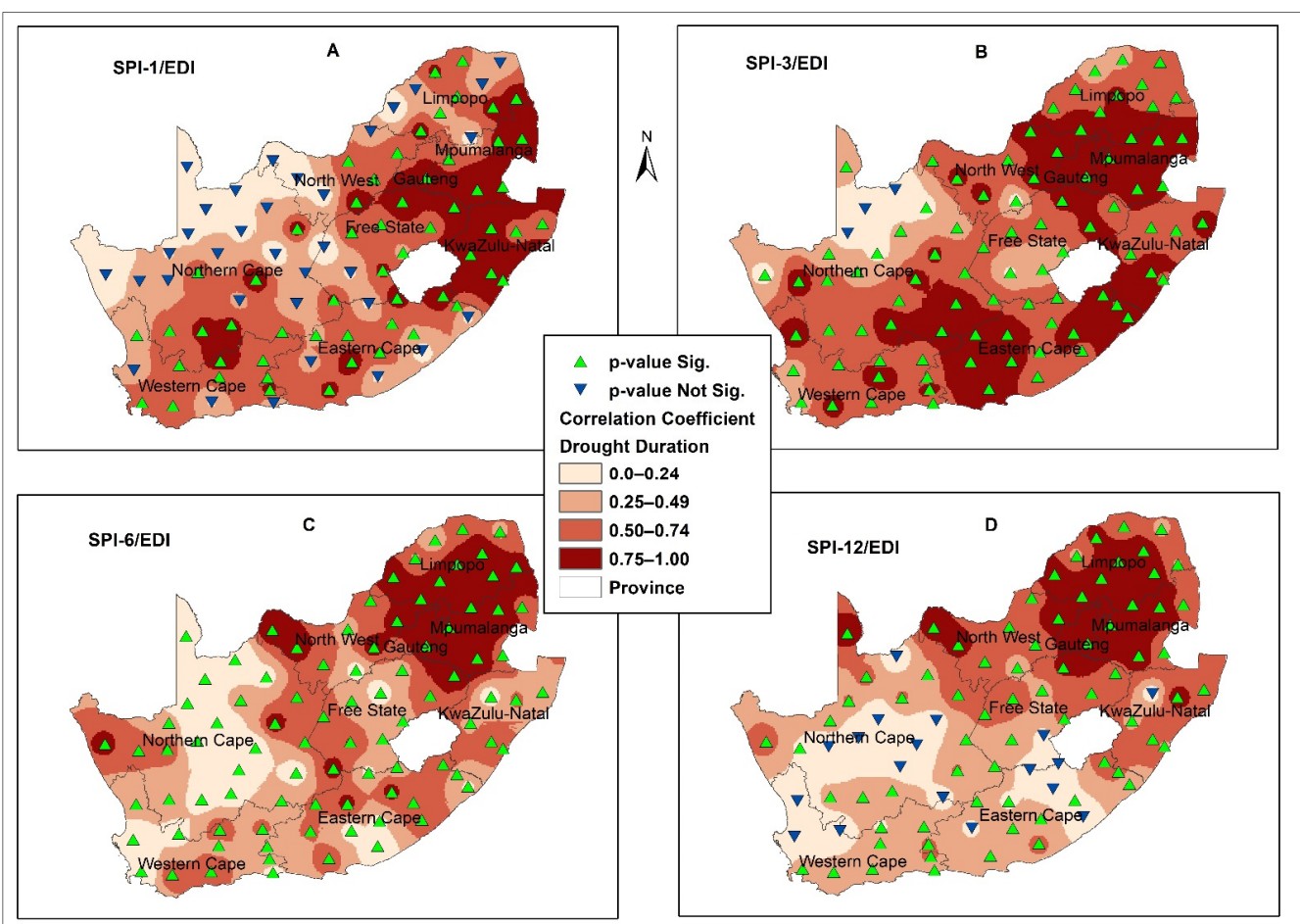

**Figure 3.** Pearson's correlation coefficient estimates dissimilarity measure for drought duration in South Africa. (**A**) SPI-1/EDI; (**B**) SPI-3/EDI; (**C**) SPI-6/EDI; and (**D**) SPI-12/EDI.

There is moderate, yet statistically significant, dissimilarity measure for 85% of the districts in the WC province for the SPI-3/EDI pair values. On the other hand, about 25% of the rainfall districts in the NC and FS provinces show weak statistically significant dissimilarity measure for the SPI-3/EDI pair, while all the rainfall districts in GT and MP provinces, as well as 70% in EC province, displays similar statistically significant values. As observed in Figures 3C and 4C, the NC province displayed strong statistically significant dissimilarity for the values of the SPI-6/EDI pair in about 55% of the province. Moderately

statistically significant dissimilarity measure was determined in 60% of the rainfall districts in NW province, and 45% of the rainfall districts for both FS and EC provinces.

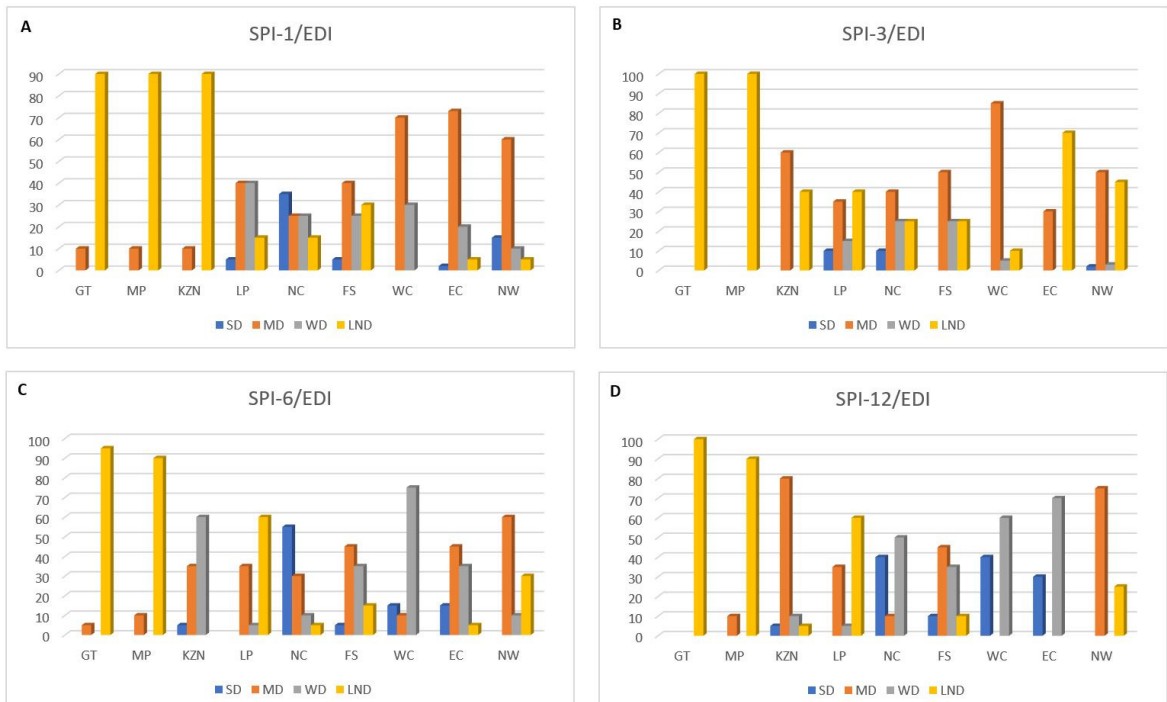

**Figure 4.** Percentage of Pearson's correlation coefficient estimate dissimilarity measure for drought duration in South Africa. (**A**) SPI-1/EDI; (**B**) SPI-3/EDI; (**C**) SPI-6/EDI; and (**D**) SPI-12/EDI. Strong Dissimilarity (SD), Moderate Dissimilarity (MD), Weak Dissimilarity (WD), Little or No Dissimilarity (LND).

Additionally, a statistically significant weak dissimilarity measures were determined for the values of the SPI-6/EDI pair in about 75% of the rainfall districts in WC province, 60% of the rainfall districts in KZN province, about 35% of the rainfall districts in both FS and EC provinces. Also, about 95%, 90%, and 60% of the districts in GT, MP, and LP provinces, respectively, exhibited similar statistically significant values for the SPI-6/EDI pair. As shown in Figures 3D and 4D, strong statistically insignificant dissimilarity measure for the SPI-12/EDI duo values were observed in 40% of the rainfall districts across both NC and WC provinces and 30% of the districts in the EC province. Moderate statistically significant dissimilarity measure for the SPI-12/EDI pair is depicted in 80% of the rainfall districts in KZN province, 75% of the rainfall districts in NW province and 45% of the rainfall districts in EC province. In EC, WC, and NC provinces, respectively, 70%, 60%, and 50% of the rainfall districts demonstrated weak, but statistically significant dissimilarity measures for the SPI-12/EDI pair. Similar statistically significant values for the SPI-12/EDI pair were found in all the rainfall districts across GT province, about 90% of the rainfall districts in MP province and 60% in LP province. The values of the metrics (the Pearson correlation's coefficients and periodogram dissimilarity measures) used to qualify (dis)similarity as Strong (SD), Moderate (MD), Weak (WD), and Little or No (LND) are given as strong if the value lies between ±0 and ±0.25, moderate if the value lies between ±0.24 and ±0.49, weak if the value lies between ±0.5 and ±0.74 and little or no if the value is between ±0.75 and ±1.

The comparison between SPI-1, SPI-3, SPI-6, SPI-12, and EDI for drought severity (DS) across the study area is portrayed in Figures 5 and 6 using the Pearson correlation coefficient estimate dissimilarity. According to Figures 5A and 6A, strong statistically insignificant dissimilarity is observed in 60% of rainfall districts in KZN province. Additionally, moderate statistically significant dissimilarity is detected in about 90% of GT province, 80% of both LP and WC provinces, and 65% of both EC and NW provinces for the SPI-1/EDI pair.

While, a weak statistically significant dissimilarity is observed in about 55%, 40%, and 40% rainfaltr45sal districts of MP, KZN, and NC provinces, respectively, for the SPI-1/EDI pair. Furthermore, the SPI-1/EDI pair shows that 50% and 20% of rainfall districts in FS and LP provinces, respectively, depicts similar statistically significant values. Figures 5B and 6B illustrate the comparison between the SPI-3/EDI pair. The figures depict a moderate statistically significant dissimilarity for the SPI-3/EDI pair is shown in 100% of the rainfall districts in both NW and GT provinces, 90% and 80% in FS and LP provinces, respectively. Whereas, a weak statistically significant dissimilarity is shown between the SPI-3/EDI in about 35% and 30% of the rainfall districts in KZN and NC provinces, respectively.

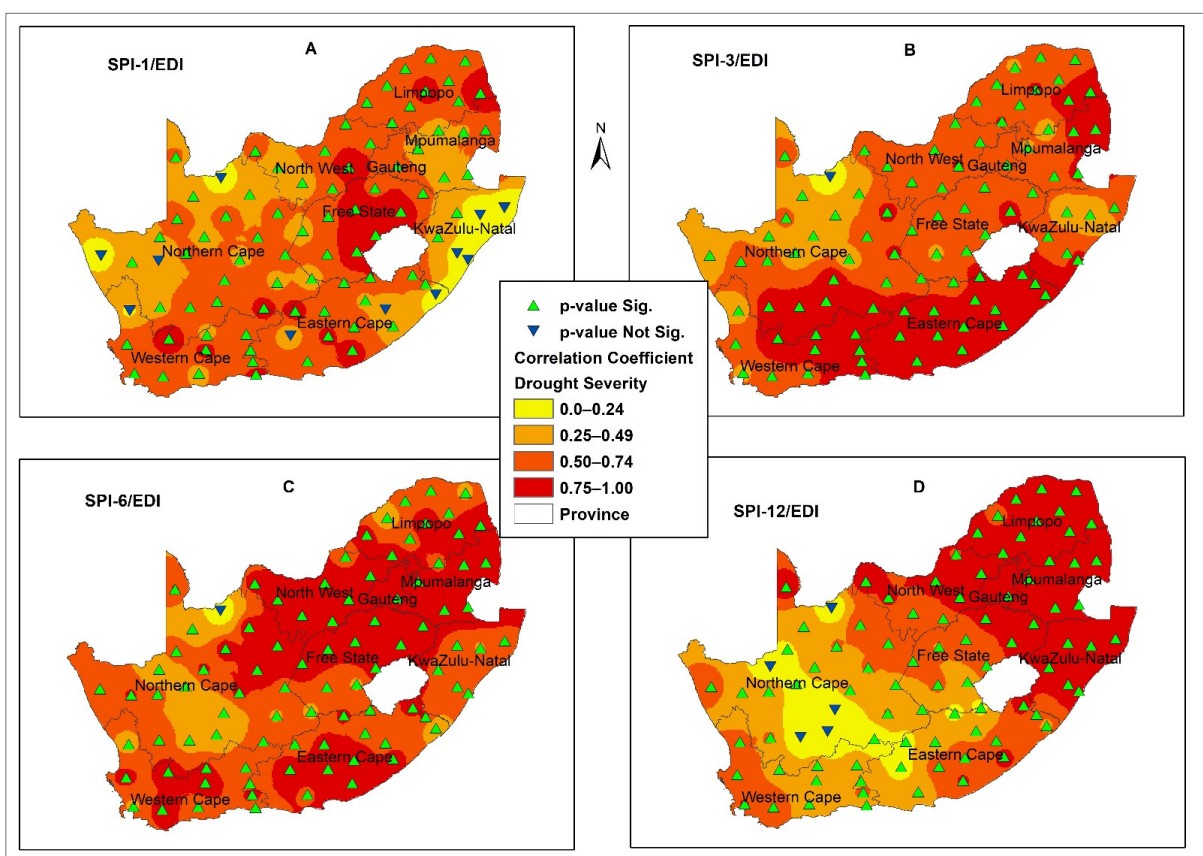

**Figure 5.** Pearson's correlation coefficient estimates dissimilarity measure for drought severity in South Africa. (**A**) SPI-1/EDI; (**B**) SPI-3/EDI; (**C**) SPI-6/EDI; and (**D**) SPI-12/EDI.

Moreover, the statistically significant similarity in 100% rainfall districts in EC province, 50% in WC, province, and 45% in MP province are also detected. In the comparison between SPI-6/EDI pair is shown in Figures 5C and 6C. The figures depict a moderate statistically significant dissimilarity is observed between the SPI-6/EDI pair, for 85%, 65%, and 53% of the rainfall districts in KZN, WC, and NC provinces, respectively, while about 30% of the rainfall districts in NC province exhibit weak statistically significant dissimilarity measure. Similar statistically significant values for all the rainfall districts in both GT and MP provinces, 95% in NW province, 65% in FS province, 62% in EC province, and 60% in LP province. On the other hand, Figures 5D and 6D illustrate the comparison between SPI-12/EDI pair. The figures show that a moderate statistically significant dissimilarity is observed between the SPI-12/EDI in about 50% of the rainfall districts in EC province, and 45% in FS province, a weak statistically significant dissimilarity is observed for about 75% of the rainfall districts in WC province, and about 40% in both EC and NC provinces. While all the rainfall districts in both GT and MP provinces, 98% in LP province, and 95% in KZN province depict statistically significant similarity measure.

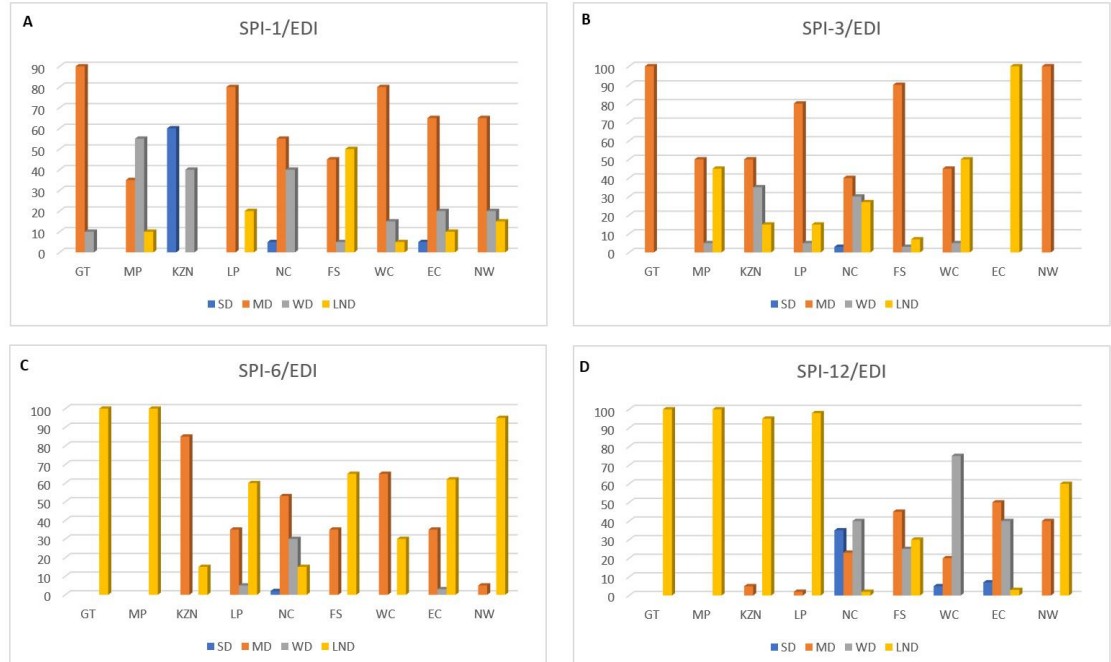

**Figure 6.** Percentage of Pearson's correlation coefficient estimate dissimilarity measure for drought severity in South Africa. (**A**) SPI-1/EDI; (**B**) SPI-3/EDI; (**C**) SPI-6/EDI; and (**D**) SPI-12/EDI. Strong Dissimilarity (SD), Moderate Dissimilarity (MD), Weak Dissimilarity (WD), Little or No Dissimilarity (LND).

### 3.2. Comparison of SPI and EDI Using Periodogram Dissimilarity

The results obtained when the periodogram measure is used for the dissimilarity analysis of DD are presented in Figures 7 and 8. The comparison for the SPI-1/EDI pair values, as shown in Figures 7A and 8A, depict strong dissimilarity, observed in about 50%, 40%, and 30% of the rainfall districts in EC, WC, and NC provinces, respectively. A moderate statistically significant dissimilarity in about 45%, 40%, and 35% of the rainfall districts in FS, GT, and NC provinces, respectively. Furthermore, the results indicate that about 60% of the rainfall districts in both MP and NW provinces and 55% in KZN province has weak dissimilarity, while similar values were detected in 15% of these provinces; LP, NC, FS, and NW. When SPI-3/EDI pair were compared, as shown in Figures 7B and 8B, strong dissimilarity were observed in about 60% of the rainfall districts in MP province, 35% in NW province, and 30% in NC, FS, and WC provinces.

Furthermore, 45% of the rainfall districts in KZN province, 40% in GT province, 30% in both LP and NC provinces exhibit moderate dissimilarity. Similarly, the SPI-3/EDI duo values show that about 60% of rainfall districts in GT, FS, and WC provinces, and about 50% of the rainfall districts in both EC and NW provinces indicate a weak dissimilarity, while similar values are observed in about 15% of the rainfall districts in LP province. The values for the SPI-6/EDI pair in Figures 7C and 8C show strong dissimilarity for 25% of the rainfall districts in MP and 30% in provinces; FS, EC, and LP. A moderate dissimilarity for about 60%, 45%, 40%, and 35% of the rainfall districts in WC, NC, FS, and NW provinces, respectively, and 90% in GT province, 60% in both MP and KZN provinces, and 50% in both EC and NW provinces present a weak dissimilarity. Similar values were observed in 20% of the rainfall districts in LP and 15% of province NC, FS, and WC. The comparison between SPI-12/EDI pair in Figures 7D and 8D, strong dissimilarity values were observed for about 45% of the rainfall districts in NC province and 50% in both KZN and NW provinces. SPI-12/EDI pair shows that about 30% of the rainfall districts in LP province and 45% in both MP and WC provinces indicate a moderate dissimilarity, while about 80%, 65%, and 45% of the rainfall districts in FS, GT, and LP provinces, respectively, suggest a weak dissimilarity. Similar values were detected in 15% of the rainfall districts in MP, LP, and WC.

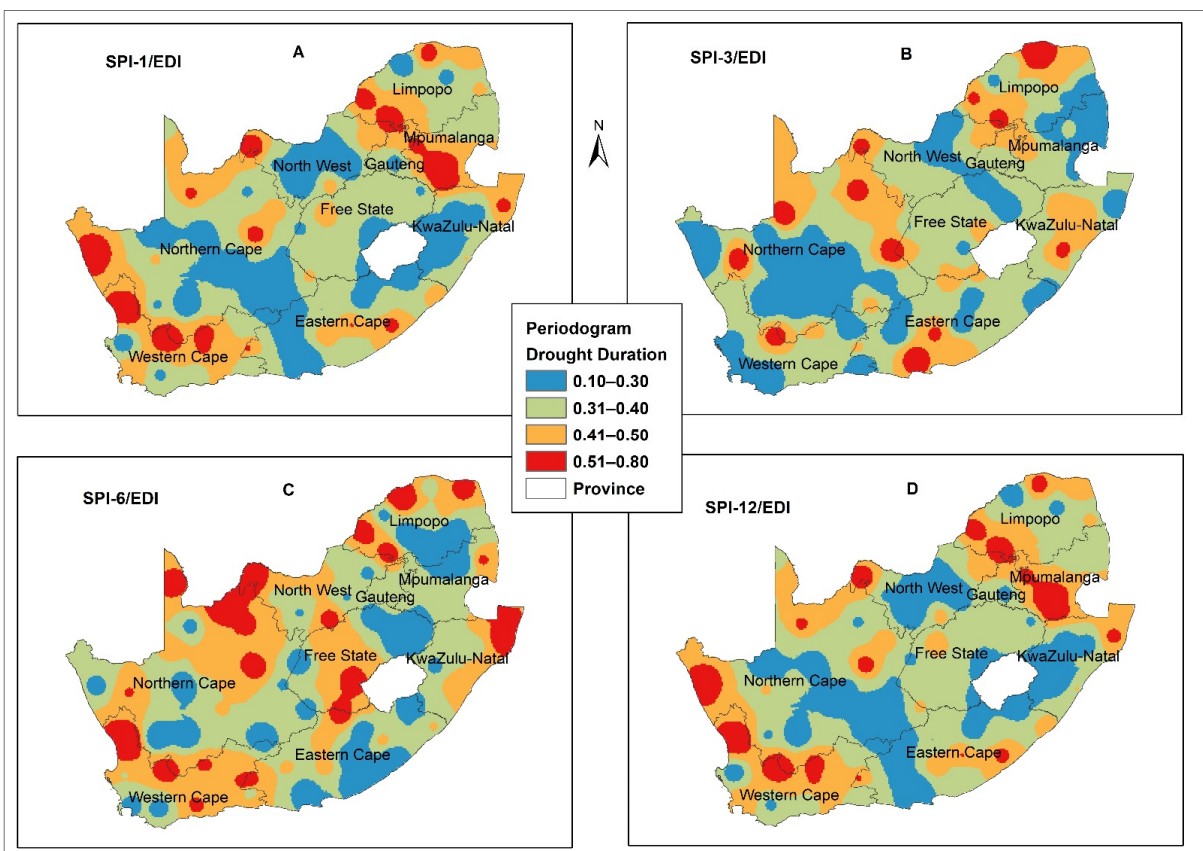

**Figure 7.** Periodogram dissimilarity measure for drought duration in South Africa. (**A**) SPI-1/EDI; (**B**) SPI-3/EDI; (**C**) SPI-6/EDI; and (**D**) SPI-12/EDI.

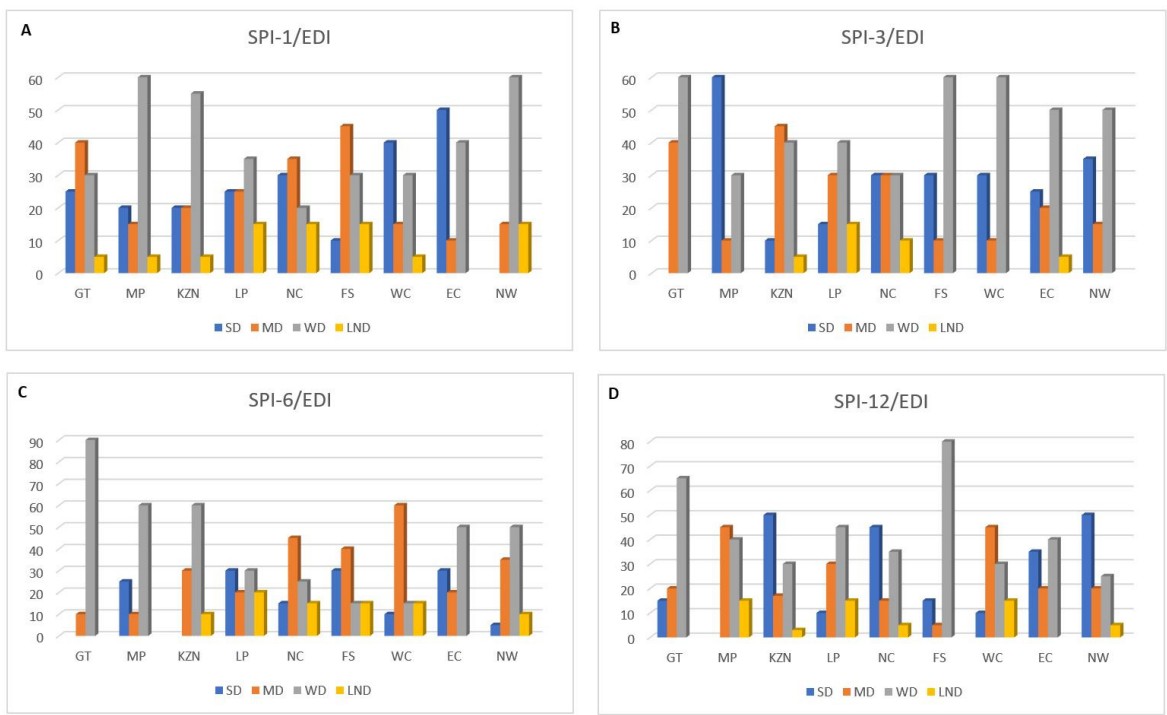

**Figure 8.** Percentage of the periodogram dissimilarity measure for drought duration in South Africa. (**A**) SPI-1/EDI; (**B**) SPI-3/EDI; (**C**) SPI-6/EDI; and (**D**) SPI-12/EDI. Strong Dissimilarity (SD), Moderate Dissimilarity (MD), Weak Dissimilarity (WD), Little or No Dissimilarity (LND).

The comparison between SPI-1, SPI-3, SPI-6, SPI-12, and EDI for drought severity over the rainfall districts of South Africa is portrayed in Figures 9 and 10 using periodogram dissimilarity. The values of the SPI-1/EDI pair (Figures 9A and 10A) show strong dissimilarity for 60%, 50%, and 30% of the rainfall districts in WC, NC, and EC province, respectively. Moreover, 90%, 70%, 60%, and 55% of the rainfall districts in GT, MP, NW, and KZN province, respectively, display a moderate dissimilarity. Additionally, SPI-1/EDI depicts that about 40% of the rainfall districts in KZN province and 50% in both FS and EC provinces display a weak dissimilarity. While similar values were observed in 30% of the rainfall districts in MP and 20% in LP. As shown in Figures 9B and 10B, a strong dissimilarity is observed in about 60% of SAWS' rainfall districts in WC province, as well as 50% of SAWS' rainfall districts in NC province for the SPI-3/EDI pair. Moreover, 60% of SAWS' rainfall districts in both GT and MP provinces, and about 50% of the rainfall districts in both FS and NW provinces show a moderate dissimilarity. Weak dissimilarity was depicted in about 65% of the rainfall districts in EC province, 45% in LP province, 35% in both KZN and NC provinces when the SPI-3/EDI pair were compared. While 40% of SAWS' rainfall districts in MP province have similar values.

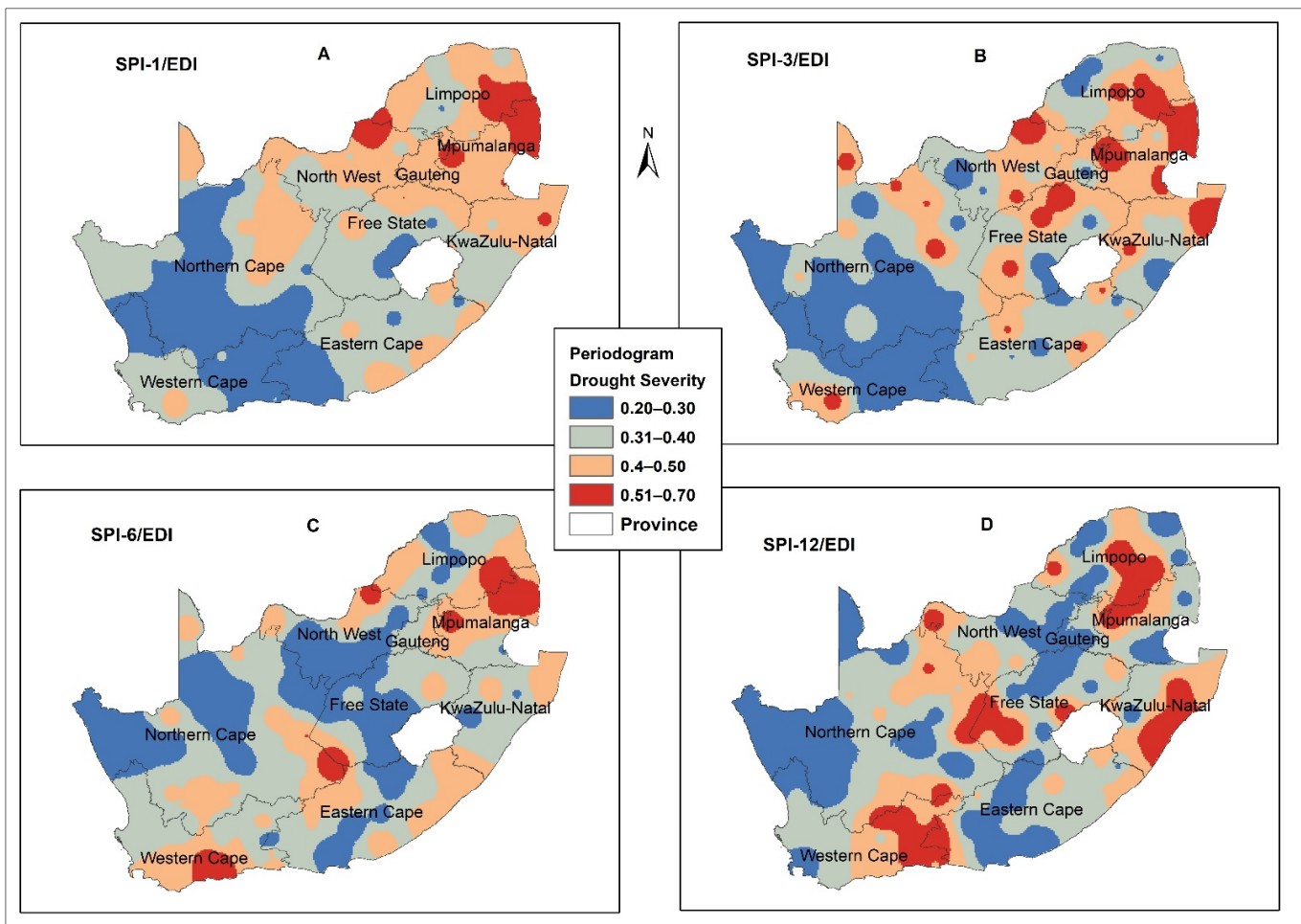

**Figure 9.** Periodogram dissimilarity measure for drought severity in South Africa. (**A**) SPI-1/EDI; (**B**) SPI-3/EDI; (**C**) SPI-6/EDI; and (**D**) SPI-12/EDI.

Similarly, the comparison between SPI-6/EDI pair (Figures 9C and 10C) depicts strong dissimilarity in 65%, 60%, and 40% of rainfall districts in NW, FS, and GT province, respectively. A moderate dissimilarity in about 60%, 45% and 40% of the rainfall districts in MP, LP, and EC province, respectively, while 65% of the rainfall districts in KZN province, 60% in both GT and WC provinces, and 48% in NC province indicate a weak dissimilarity.

In addition, the SPI-6/EDI pair displays similar values in 25% of the rainfall districts in MP province. When SPI-12/EDI pair values (Figures 9D and 10D) were compared, the results show that strong dissimilarity was detected for 40% of the rainfall districts in both GT and NC provinces, 35% in EC province, and 30% in both MP and NW provinces. Moreover, 40% and 30% of the rainfall districts in KZN and WC province, respectively, exhibit a moderate dissimilarity, while 60% of the rainfall districts in GT province, 45% in EC province, 40% in MP province, NC province, and NW province show a weak dissimilarity. Furthermore, the SPI-12/EDI duo values display similarity for 25% of the rainfall districts in LP province and 20% in both KZN and WC provinces.

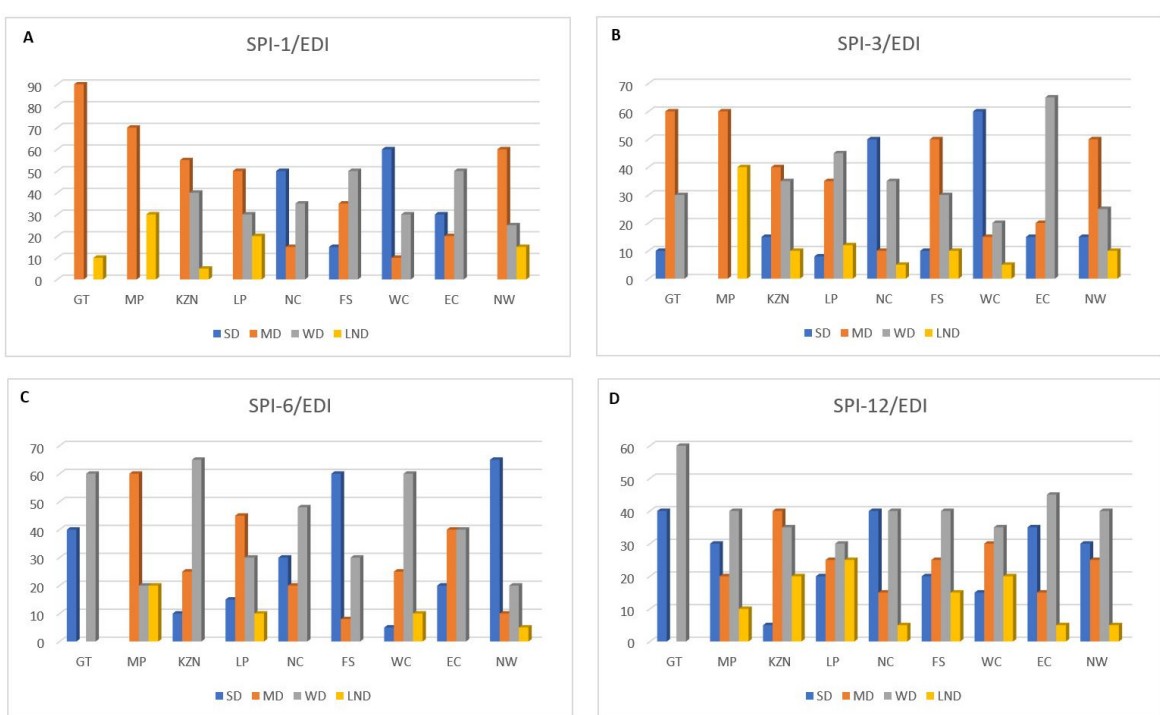

**Figure 10.** Percentage of the periodogram dissimilarity measure for drought severity in South Africa. (**A**) SPI-1/EDI; (**B**) SPI-3/EDI; (**C**) SPI-6/EDI; and (**D**) SPI-12/EDI. Strong Dissimilarity (SD), Moderate Dissimilarity (MD), Weak Dissimilarity (WD), Little or No Dissimilarity (LND).

## 4. Discussion

Frank et al. [4] made remarks that EDI has good similarity with the different SPI accumulation periods, which makes it capable of recognizing all droughts. In other words, it can recognize drought of different duration, giving a single value. This is evident in this study, given the similarity observed in virtually all the scenarios. A similarity measure provides a score that defines how similar a feature of the two vectors is, divergence from distances, and dissimilarity measures, which give a score relating how much two items differ. This study investigates the similarity and dissimilarity between the two frequently used indices (SPI and EDI) for drought duration (DD) and severity (DS). The SPI-1, SPI-3, SPI-6, and SPI-12 is compared with the EDI values using the Pearson correlation coefficient estimate dissimilarity and periodogram measures. The study is conducted across the 93 South African Weather Service's delineated rainfall districts over South Africa from 1980 to 2019.

The comparison between the SPIs (that is, SPI-1, SPI-3, SPI-6, and SPI-12) and EDI shows variations in the dissimilarity and similarity among these drought indicators. However, there were no negative values for both the Pearson correlation coefficient estimate dissimilarity and the periodogram dissimilarity. Comparison with the Pearson correlation coefficient estimate dissimilarity shows that the drought indices/indicators for drought

duration and severity displays similarity in quite several districts of the study area, some of these districts are located in the GT and MP provinces. For instance, in the comparison between SPI-3/EDI pair for DD, 100% of SAWS' rainfall districts in GT and MP province exhibit noticeable similarity, which is statistically significant—this implies that SPI-3/EDI values can be used interchangeably in both GT and MP provinces. The majority of the districts depict moderate dissimilarity among the drought indices for DD and DS, illustrating spatial contrasts of the SPI/EDI values across South Africa. Except for the comparison between SPI-12/EDI pair, where the majority of the stations depict weak dissimilarities for the drought duration and drought severity. The similarities between SP1-1, SPI-3, SPI-6, SPI-12, and EDI were more evident in the northern region of the study area for DD. Similarly, for DS, the similarities were predominantly in the northern region except for the comparison between SPI-3/EDI pair. Also, the study shows that the values of the SPI-1/EDI pair and the SPI-3/EDI pair exhibit the highest similar values for DD, while SPI-6/EDI pair shows the highest similar values for DS. Dissimilarities are more evident in SPI-12/EDI pair for DD and DS. These results corroborate the findings of [4], which state that various values of SPIs are replaceable with one value of EDI. In this study, the SPI-1 values can be replaceable with the EDI values for DD, and the SPI-6 values can be replaceable with the EDI values for DS in South Africa.

The case is fairly different with periodogram dissimilarity as stations with similar values are more, cutting across all the provinces. Many of the districts, has weak dissimilarity among the drought indices for drought duration and drought severity, except for the comparison between SPI-1/EDI for DS, where moderate dissimilarity is predominant. Furthermore, for the drought duration, the similarities were obvious in the western interior and the northern part of South Africa for the SPI-1, SPI-6, and EDI pairs, while it was predominant in the northern and southern part of South Africa for the SPI-12/EDI pair. The similarities were more evident in the northern part of South Africa for the SPI-1/EDI pair, the northern and western part of South Africa for the SPI-3/EDI pair, the northern and southern part of South Africa for the SPI-6/EDI pair and SPI-12/EDI pair for DS. Additionally, the values of the SPI-1/EDI pair and SPI-6/EDI pair exhibit the highest similar values for DD, while SPI-1/EDI displayed the highest similar values for DS. Moreover, drought duration and severity show that similarities are obvious in SPI-1/EDI. This study indicates that in South Africa, the SPI-1 and SPI-3 values can be used interchangeably with the EDI values for DD, while for DS, the SPI-1 values can be interchangeably used with the EDI values. In general, the results show variation in the similarity and dissimilarities of the indices across the entire study area. The observed differences in the (di)similitudes could be related to regional variation in climatic conditions across the regions, as reported in Reference [49].

## 5. Conclusions

Studies have demonstrated the application of 1- to 3-month accumulation of SPI for short-term precipitation anomalies, such as meteorological drought, agricultural drought, and soil moisture retention. Whereas, the 6- to 12-month accumulation of SPI timescales have been used for assessing drought impact on reservoir levels, groundwater, and streamflow. Based on the similarity in mathematical computation of the EDI and SPI, it provides an advantage for comparing the two indices for drought determination. In this study, the Pearson's correlation coefficient dissimilarity measure with *p*-values and the periodogram dissimilarity measure was used to evaluate the similarity and dissimilarity between the two indices; EDI and SPI accumulation periods of 1-, 3-, 6-, and 12-months. The study provides insight into the variation in the performance, sensitivity, and suitability of drought indices across the South African landscape. The results further illustrate the similarity and dissimilarity between the SPI accumulation periods of 1-, 3-, 6-, and 12-months and EDI. Consequently, the overall results deduce from this study can be summarized as follows:

The comparison between SPI and EDI in the study area shows dissimilarities; however, the difference was moderate, which implies that the SPI and EDI values cannot be used

interchangeably in monitoring drought for South Africa. As illustrated in the results, the similarity is predominantly in the northern region of the study area.

The two measures show that the comparison between SPI-1/EDI has the highest similarity for drought severity. This implies for stations where high similarity is found, SPI and EDI values can be used interchangeably when considering the drought severity for meteorological drought, agricultural drought, and soil moisture retention. Furthermore, the two measures show that the comparison between SPI-12/EDI has the highest dissimilarities for drought duration for drought impact on reservoir levels, groundwater, and streamflow.

Given the strong similarity between EDI and SPI-1, the results suggest that the two indices can be used interchangeably for meteorological and agricultural drought monitoring over the study area. This study further supports the WMO's recommendation of SPI for drought monitoring.

Drought periodicity is highly complex—there is, however, the need to effectively monitor drought to enhance proper planning to mitigate the effect of drought in South Africa. This contribution enlightens the fact that SPI and EDI values are replaceable in some rainfall districts of South Africa for drought monitoring and prediction, reducing the rigor of calculating the other drought indices in situations where one of the drought indices is available. Furthermore, studies involving the use of historical records alongside the SPI and EDI to determine the indices that best detect South African drought is recommended.

**Author Contributions:** O.M.A. conceived the research. O.M.A. and J.O.B. analyzed the data and drafted the paper. M.M. reviewed the manuscript. All authors have read and agreed to the published version of the manuscript.

**Funding:** The project reported in this paper is partially funded by SIDA, through the Swedish International Development Cooperation Agency, the Foreign Ministry of the Netherlands, the South African Department of Science and Technology, and USAID under Award No. AID-OAA-F-17-00034 Under Securing Water for Food: A Grand Challenge for Development FRA Number: SOL-OAA-16-000176. The project is also partially funded by the South Africa Research Foundation (NRF) grant for 2019: Thuthuka Funding Instrument (Unique Grant No: 117800). This research is also supported by the African Institute for Mathematical Sciences, www.nexteinstein.org, with financial support from the Government of Canada, provided through Global Affairs Canada, www.international.gc.ca, and the International Development Research Centre, www.idrc.ca.

**Institutional Review Board Statement:** The study does not require ethical approval.

**Informed Consent Statement:** Not applicable.

**Data Availability Statement:** Data used for this study can be made available upon request.

**Acknowledgments:** The authors gratefully acknowledge financial support from the Department of Science and Technology (DST) South African Weather Services (SAWS) and The National Research Foundation (NRF).

**Conflicts of Interest:** The authors declare no conflict of interest.

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
