# Peer review of "Assessment of the Dissimilarities of EDI and SPI Measures for Drought Determination in South Africa"

_water, doi:10.3390/w13010082_

Round 1

Reviewer 1 Report

Even if the paper addresses an interesting topic, there are some issues that need to be strengthened on the paper, namely:

Literature review misses significant works and important advances regarding drought analysis.

Material and methods are hard to read and understand. The used methodology and de established research steps are not adequately described. I would recommend the introduction of a phased methodological diagram.

The results are fine.

Conclusions need to be more specific and more scientific. As they are they highlight the limitations of the research.

Author Response

The authors have addressed the comments made by the reviewers. All the comments and suggestions made by the reviewer have been taken into consideration. Consequently, we are glad to send you a much improved manuscript.

Details of our responses are contained in the attached document. 

Reviewer 2 Report

It is widely recognized the necessity to shift from a reactive to a proactive drought risk management to reduce the impact of droughts on natural resources and human livelihoods. This new approach will require of the development of early warning systems built on  drought indexes for monitoring the occurrence, duration and severity of droughts. In this regards, this paper presents an analysis of the (dis)similitudes between two commonly used drought indices across South Africa in order to find where the indexes are replaceable each other and make recommendations on which is best fitted for use. The topic is certainly of the interest and within the scope of the journal Water. The manuscript has, however, some important drawbacks that lead me not to recommend the publication at its current form and suggest the authors resubmitting a revised version.

In its current form, the paper only provides a thorough description of the dissimilarities found between the two indexes: SPI and DSI by applying two different metrics: the Pearson correlation coefficient and the periodogram. Besides this descriptive information, the authors should have provided a deeper discussion on the results obtained and a more conclusive recommendation on selecting drought indices.

There are several aspects I would have like to have been discussed in the paper:

The effects of the accumulation period in the calculation of the SPI on the (di)similitudes with EDI. Did these differences rise because of the conceptual basis of the EDI, i.e. the cumulative effect of precipitation deficit on the value of EDI?

Are SPI and EDI more, or less, interchangeable when monitoring drought duration or severity?. If so, why?

Which are the reasons of the differences observed in the (di)similitudes across regions? Are they related to regional variation in climatic conditions or in any other variables?

Why were two methods used to calculate the (di)similarities? Which are the pro and cons of each of the methods? Did the authors reach any conclusion about which is the best suitable method?

Given that the WMO recommended SPI as the main meteorological drought index that countries should use to monitor and follow drought conditions (Hayes, 2011), it would be expected that authors discussed about the suitability of the indexes for drought monitoring and risk management and the potential advantages of using EDI instead of SPI

In drafting the revised version the authors may also want to consider other minor comments

Abstract should be shortened  focusing on the more relevant rather than on an exhaustive description of the results. It should also refer to main conclusions in terms of exchangeability between indices and how well they are fitted for drought monitoring.

On methods, readers are very often asked to look for in original bibliographic sources so the description is not detailed enough for a good understanding. In particular more information is needed for periodogram based dissimilarity measure (line 192)

I hesitate about having well understood how dissimilarity is measured. For instance, in figure 2 darker brown areas with higher values of Pearson coefficient are reported as zones with greater dissimilarities. To me, the Pearson correlation coefficient reflects the linkages between two variables, that means that as close to 1 the coefficient is as more linked the two variables are.  Would you please clarify it?

In figures 3 and 5: what do SD,WD,LND and MD stand for?

How were drought duration (DD) and severity (DS) defined and calculated?. Please clarify it.

References

Hayes, M., M. Svoboda, N. Wall and M. Widhalm, 2011: The Lincoln Declaration on Drought Indices: Universal Meteorological Drought Index Recommended. Bulletin of the American Meteorological Society, 92(4): 485–488. DOI: 10.1175/2010BAMS3103.1.

Author Response

(The authors gave the same response as above.)

Reviewer 3 Report

All in all I found your study well-designed, concisely written and interesting for the community. Unfortunately, alone the Introduction is of rather underground quality, presumably owed to your skills in mathematical statistics more than drought assessment and evaluation. Please, rewrite this chapter from scratch - after having a close look at my comments and remarks. You may find them in the annotated PDF attached. 

Author Response

(The authors gave the same response as above.)

Round 2

Reviewer 2 Report

I acknowledge the effort made by the authors to respond to the comments and request made by the reviewers in the previous round and to draft an improved version of the manuscript.

Despite of this improvement, the manuscript still requires further work before it is eventually accepted for publication. Some points that can be considered by the authors when drafting a revised version are as follow:

In the introduction, the authors state that (i) “there is insufficient knowledge on the sensitivity of the drought monitoring indices across national territory (line, 108); and (ii) “there is a need to understand the advantages and disadvantages of the several drought indices” (line112).  Both questions are pertinent and relevant, so I would recommend the authors to clearly show how this study addresses these research questions and the conclusions reached In line 102, the authors mention that the index EDI is recommended for effective identification of drought onset, end and duration ; by who?. Please provide references

 As indicated in my previous review, further details on how to calculate periodogram dissimilarity measures are needed so it becomes understandable for readers who are not acquainted about statistical time series analysis.

Line 212 says “From Figure 3A and 4A, strong statistically insignificant dissimilarity between SPI-1/EDI pair can be observed in 35% of the rainfall districts in NC province”. what do the authors really want to say by strong insignificant dissimilarity? . When showing the results the authors should consider to clearly distinguishing between statistical significance of the (dis)similitude and their strength (from strong to weak).

The range of the values of the metrics (Pearson correlation´s coefficients and periodogram dissimilarity measures) used to qualify (dis)similarity as SD, MD, WD and LND should be provided.

Can the authors provide any explanation about why  (dis)similarities found  between drought indexes are different depending on the method, Pearson correlation coefficient or periodogram (dis)similarities measures, used?

Lines 398-400 seem to be unfinished and further elaboration is required

Reviewer 3 Report

Well done, thank you.